# ADPN Regulates Oxidative Stress-Induced Follicular Atresia in Geese by Modulating Granulosa Cell Apoptosis and Autophagy

**DOI:** 10.3390/ijms25105400

**Published:** 2024-05-15

**Authors:** Yan Zheng, Yunqiao Qiu, Qianhui Wang, Ming Gao, Zhongzan Cao, Xinhong Luan

**Affiliations:** College of Animal Science and Veterinary Medicine, Shenyang Agricultural University, Shenyang 110866, China; 2023200184@stu.syau.edu.cn (Y.Z.); 2021220584@stu.syau.edu.cn (Y.Q.); 2023220613@stu.syau.edu.cn (Q.W.); 2003500039@syau.edu.cn (M.G.)

**Keywords:** goose, adiponectin, granulosa cell, oxidative stress, apoptosis, autophagy, follicular atresia

## Abstract

Geese are susceptible to oxidative stress during reproduction, which can lead to follicular atresia and impact egg production. Follicular atresia is directly triggered by the apoptosis and autophagy of granulosa cells (GCs). Adiponectin (ADPN), which is secreted by adipose tissue, has good antioxidant and anti-apoptotic capacity, but its role in regulating the apoptosis of GCs in geese is unclear. To investigate this, this study examined the levels of oxidative stress, apoptosis, and autophagy in follicular tissues and GCs using RT-qPCR, Western blotting, immunofluorescence, flow cytometry, transcriptomics and other methods. Atretic follicles exhibited high levels of oxidative stress and apoptosis, and autophagic flux was obstructed. Stimulating GCs with H_2_O_2_ produced results similar to those of atretic follicles. The effects of ADPN overexpression and knockdown on oxidative stress, apoptosis and autophagy in GCs were investigated. ADPN was found to modulate autophagy and reduced oxidative stress and apoptosis in GCs, in addition to protecting them from H_2_O_2_-induced damage. These results may provide a reasonable reference for improving egg-laying performance of geese.

## 1. Introduction

Follicular development plays a crucial role in determining reproductive performance in animals. In poultry, follicle production and development occur continuously throughout the reproductive period. The follicles form a hierarchical system in the ovary and develop sequentially in a set order [1]. Follicles are typically classified into two categories based on their size and functional status: pre-hierarchical follicles and hierarchical follicles. Pre-hierarchical follicles consist of small white follicles (SWFs), large white follicles (LWFs), small yellow follicles (SYFs), and large yellow follicles (LYFs). The hierarchical follicles are named F1–F5 in order of size [2,3]. Primordial follicles have three fates: (1) they remain dormant as part of the follicular pool; (2) they are activated by reproductive hormones, growth factors, and other factors, and gradually mature; or (3) they deteriorate and die while in the dormant state due to various factors [4]. In avian species, the primordial follicles that develop into SYFs are the only ones recruited for further progression, and then selection occurs and they gradually develop into hierarchical follicles that ovulate sequentially from F1 to F5 [5]. However, only a very small percentage of primordial follicles in both mammalian and avian species are able to mature and ovulate, and the vast majority of follicles undergo follicular atresia during development.

Granulosa cells (GCs) synthesize a variety of steroid hormones and growth factors, which regulate the growth and differentiation of follicular membrane cells and oocytes, and thus follicular development. The growth and differentiation of granulosa cells play an important role in primordial follicle activation and subsequent follicle selection and development. Therefore, the growth and differentiation of granulosa cells can be used as a marker of follicle development [6]. Follicular atresia is triggered by the death of GCs, mainly through cell apoptosis and autophagy. Previous studies showed that the apoptosis of GCs can cause follicular atresia, a phenomenon observed in mice, chickens and geese [5,7,8]. However, it is important to note that follicular atresia is not solely controlled by cell apoptosis, as enhanced autophagy of GCs has also been observed during follicular atresia in mice and geese [9,10].

During the breeding process, oxidative stress can occur in poultry, which produces reactive oxygen species (ROS) and damages the follicular microenvironment. Excessive accumulation of ROS in follicular fluid can lead to the apoptosis and autophagy of GCs, ultimately resulting in follicular atresia [11,12,13]. Other evidence also suggests that ROS can cause apoptosis and autophagy in poultry follicular GCs [10,14,15].

Adiponectin (ADPN) is a cytokine secreted by adipose tissue. It regulates glucose and lipid metabolism, insulin sensitivity, and inflammation resistance by binding to its receptor [16,17]. ADPN and its receptors are present in the follicular GCs of humans, mice, and poultry [18,19]. Evidence suggested that ADPN has strong anti-oxidative stress and anti-apoptotic capabilities. ADPN can reduce ROS and malondialdehyde (MDA) levels, increase superoxide dismutase (SOD) and glutathione peroxidase (GSH-Px) activities, increase B-cell lymphoma-2 (Bcl-2) expression, decrease BCL2-Associated X (Bax) expression, and prevent ROS-induced decreases in mitochondrial membrane potential [20,21].

Disruption of the ADPN receptor, AdipoR1, results in apoptosis in the granulosa cell line KGN [22]. Furthermore, high levels of ADPN contribute to the protection of GCs from apoptosis [23]. Our previous study showed that ADPN can regulate the secretion of steroid hormones to modulate reproductive activity in geese [24]. In this study, the effects of oxidative stress, apoptosis, autophagy and the expression of adiponectin were investigated in follicles from geese at different stages of development, and atretic follicles were also investigated. This study also sought to ascertain whether the overexpression or knockdown of ADPN could regulate apoptosis and autophagy of goose primary GCs caused by H_2_O_2_ through in vitro analysis. Our findings might provide a theoretical basis and reasonable therapeutic strategies for improving egg production rates and treating related diseases in geese.

## 2. Results

### 2.1. Morphology of Follicles at Different Stages of Development and Histological Characteristics of Each Membrane Layer of Follicles in Geese

The morphological features of follicles at different stages were observed, in particular changes in the color and elasticity of atretic follicles and the loss of blood vessels on the surface of the membrane (Figure 1A). The histological characteristics of each membrane layer of follicles at different stages were observed, and the results showed that the layers of the AFs were loosely arranged and that the GC layer was detached (Figure 1B).

### 2.2. High Levels of Oxidative Stress in Pre-Hierarchical and Atretic Follicles

The antioxidant capacity of atretic follicles was found to be reduced, with significantly lower SOD viability compared to hierarchical follicles (Figure 2A). Additionally, the MDA content was significantly higher in atretic follicles compared to hierarchical follicles. LYFs and SYFs exhibited a comparable pattern to atretic follicles: they had significantly higher MDA levels compared to hierarchical follicles. (Figure 2B), possibly due to being in an early stage of follicular development, where the antioxidant capacity is weaker and the follicles are more vulnerable to oxidative stress.

### 2.3. High Levels of Apoptosis in Atretic Follicles

To determine the level of apoptosis in follicles at each developmental stage, the mRNA and protein expression of apoptosis-related genes was determined. There were higher levels of apoptosis in atretic follicles, with increased expression of the pro-apoptotic genes *P53*, *caspase-3* and *Bax* mRNA and decreased expression of the anti-apoptotic gene *Bcl-2* (Figure 3A), a significant increase in Bax protein expression (Figure 3B,C), a significant decrease in Bcl-2 protein expression (Figure 3B,D), and a decrease in the ratio of Bcl-2 to Bax (Figure 3E). We also found higher levels of apoptosis in F1 follicles. These results indicated that atretic follicles had a higher level of apoptosis and that F1 follicles might be transitioning progressively toward atresia.

### 2.4. Autophagy Is Involved in Follicular Atresia

Transmission electron microscopy was used to observe the submicroscopic structure of atretic follicles. The atretic follicles showed nucleolysis and the loss of mitochondrial cristae. Additionally, mitochondria with disappearing cristae were observed to be phagocytosed by autophagosomes (Figure 4A).

The mRNA and protein expression of autophagy-related genes were investigated. The expression levels of *ATG5*, *ATG7*, *Beclin1*, *P62* and *LC3* were higher in atretic follicles compared to hierarchical follicles. The same trend was observed in pre-hierarchical follicles, particularly SYFs, although statistical significance was not achieved for some of these changes (Figure 4B). Similar to their mRNA expression levels, the expression of P62 and LC3-II proteins, as well as the ratio of LC3-II to LC3-I protein expression, were higher in atretic follicles and SYFs compared to hierarchical follicles (Figure 4C–G). These data suggest that autophagy plays an important role in the process of follicular atresia.

### 2.5. Expression of ADPN in Goose Follicles at Different Stages

Upon examining the mRNA and protein expression of ADPN, it was discovered that *ADPN* mRNA expression was lower in atretic follicles compared to hierarchical follicles, and its expression decreased progressively as the follicles developed into F1 follicles (Figure 5A). However, there was no significant difference in the level of ADPN protein among the follicles (Figure 5B,C).

### 2.6. Oxidative Stress Induces Apoptosis and Autophagy in Goose GCs

To determine the effect of oxidative stress on GCs, primary GCs were isolated (Figure 6A), and FSHR-positive expression was used as a criterion for identification (Figure 6B). An oxidative stress model of GCs was constructed by stimulating GCs with 250 μmol/L H_2_O_2_ for 6 h, based on previous experiments conducted in our laboratory. Viability of the GCs was detected using a CCK8 assay and via observation of their morphology. GC viability was significantly reduced in the H_2_O_2_ group compared to the control group (Figure 6C). Furthermore, after 6 h of H_2_O_2_ stimulation, the morphology of the GCs was altered, and intercellular junctions were disrupted (Figure 6D). The ROS assay results suggested that H_2_O_2_ stimulation induced oxidative stress in GCs (Figure 6E). The apoptosis of GCs was examined, revealing increased expression of the pro-apoptotic genes *P53*, *Caspase-3* and *Bax* and decreased expression of the anti-apoptotic gene *Bcl-2* in the H_2_O_2_ group compared to the control group (Figure 7A). High levels of apoptosis were similarly revealed in the H_2_O_2_ group through Hoechst33342/PI staining (Figure 7B) and immunofluorescence (Figure 7C) results. Additionally, the increased apoptosis rate of GCs in response to H_2_O_2_ stimulation was confirmed by means of flow cytometry (Figure 7D). Similarly, the level of autophagy in GCs was assessed, and it was found that the expression levels of *ATG5*, *ATG7*, *Beclin 1*, *P62* and *LC3* all increased in response to H_2_O_2_ stimulation (Figure 7E). The expression of the Beclin 1 protein was also increased, as determined by means of immunofluorescence (Figure 7F). These findings suggested that oxidative stress induced apoptosis and autophagy in GCs by increasing ROS production.

### 2.7. ADPN Regulated Apoptosis and Autophagy in GCs Induced by Oxidative Stress in GCs

To assess the effect of ADPN on apoptosis and autophagy in GCs under oxidative stress, specific siRNA and overexpression plasmids were used to decrease or increase its expression, respectively. The mRNA expression of *ADPN* was reduced in GCs transfected with siRNA-ADPN (siADPN) compared to the negative control group (NC), and the mRNA expression of *ADPN* increased in GCs transfected with the ADPN overexpression plasmid (ADPN O/E) compared to the vector group (Figure 8A). The same trend was observed for ADPN secretion in the culture supernatant of GCs (Figure 8B). The ROS assays results suggested that ADPN has a regulatory effect on oxidative stress in GCs. The knockdown of ADPN resulted in an increase in ROS production, which further enhanced H_2_O_2_-induced ROS production. Conversely, the overexpression of ADPN led to a decrease in ROS production (Figure 8C).

The downregulation of ADPN induced apoptosis in GCs. The data indicated an increase in the expression of *P53*, *Caspase-3*, and *Bax*, a decrease in the expression of *Bcl-2* and a decrease in the *Bcl-2*/*Bax* ratio (Figure 9A–E). Additionally, the number of Hoechst33342/PI-staining-positive cells (Figure 10A), the level of Caspase-3 protein expression (Figure 10B) and the apoptosis rate (Figure 10C,D) all increased after ADPN knockdown. This trend was more noticeable when siADPN and H_2_O_2_ were combined to influence GCs. In contrast, upregulation of ADPN resisted apoptosis. Compared to the vector group, the ADPN O/E group exhibited a decrease in *P53*, *Caspase-3* and *Bax* expression, an increase in *Bcl-2* expression and an increase the *Bcl-2*/*Bax* ratio. (Figure 9A–E). Furthermore, the number of Hoechst33342/PI-staining-positive cells (Figure 10A), the level of Caspase-3 protein expression (Figure 10B) and the apoptosis rate (Figure 10C,D) were reduced while attenuating H_2_O_2_-induced apoptosis.

ADPN regulates cellular autophagy. The knockdown of ADPN inhibits autophagy, while the overexpression of ADPN promotes it. The siADPN group exhibited decreased mRNA expression of *ATG5*, *ATG7*, *Beclin1* and *LC3*, as well as increased expression of *P62* mRNA (Figure 11A–E), and decreased protein expression of Beclin1 (Figure 11F). In contrast, the ADPN overexpression group showed increased mRNA expression of *ATG5*, *ATG7*, *Beclin1* and *LC3*, decreased *P62* mRNA expression and increased Beclin1 protein expression. Stimulating GCs with H_2_O_2_ led to a decrease in mRNA expression of *ATG5*, *ATG7*, *Beclin 1*, and *LC3*, as well as a decrease in Beclin 1 protein expression in both the siADPN + H_2_O_2_ group and the ADPN O/E + H_2_O_2_ group. Furthermore, *p62* expression was increased in the siADPN + H_2_O_2_ group and decreased in the ADPN O/E + H_2_O_2_ group (Figure 11A–F).

Since the biological effects of ADPN are mediated by its binding to the AdipoR1 and AdipoR2 receptors, the expression levels of AdipoR1 and AdipoR2 were next examined. The results showed that both *AdipoR1* and *AdipoR2* mRNA expression increased (Figure 12A,B), AdipoR1 protein expression level was significantly elevated, and there was no significant difference in AdipoR2 protein expression level in the ADPN O/E + H_2_O_2_ group compared to the H_2_O_2_ group (Figure 12C). There was no significant difference in both AdipoR1 and AdipoR2 mRNA and protein expression in the siADPN + H_2_O_2_ group compared to the H_2_O_2_ group (Figure 12A–C).

To investigate the role of ADPN in regulating apoptosis and autophagy, RNA-seq analysis was conducted on each group of GCs. The differentially expressed genes (DEGs) in each group of GCs are shown in Figure 13A. A total of 723 DEGs were identified between the control group and the H_2_O_2_ group, of which 477 were upregulated and 246 were downregulated; 75 DEGs were identified between the NC group and siADPN group, of which 58 were upregulated and 17 were downregulated; 106 DEGs were identified between the vector group and ADPN O/E group, of which 25 were upregulated and 81 were downregulated; 112 DEGs were identified between the siADPN group and siADPN + H_2_O_2_ group, of which 77 were upregulated and 35 were downregulated; and 88 DEGs were identified between the ADPN O/E group and ADPN O/E + H_2_O_2_ group, of which 31 were upregulated and 57 were downregulated.

Furthermore, GO analysis of DEGs in the control and H_2_O_2_ groups enriched biological processes such as regulation of the apoptotic progress (DEGs: *MCL1*, *FGF10*, *MAPK7*, etc.), cell cycle (DEGs: *CAPN3*, etc.), positive regulation of oxidative-stress-induced cell death (DEGs: *MCL1*, etc.), and inflammatory response (DEGs: *CTSS*, *NFκBIA*, *CCR2*, *CCR7*, etc.), negatively regulating of cell growth (DEGs: *SRF*, *PAK1*, *MAP3K9*, etc.), differentiation (DEGs: *ESRRβ*, *HOXC9*, etc.), immune response (DEGs: *CRK*, *ACOD1*, *ADRA2B*, etc.), post-embryonic development (DEGs: *APC*, *SMARCC2*, etc.), etc. (Figure 13B). The GO analysis also showed the NC and siADPN groups to be enriched in DEGs negatively regulating the cell cycle (DEGs: *PLK2*, *PCNA*, *CDKN2B*, etc.) and immune response (DEGs: *TGFβ3*, *SAMSN1*, etc.) and positively regulating DNA damage (DEGs: *FGF10*, *PCNA*, *CDKN2B*, etc.), inflammatory responses (DEGs: *AGTR1*, *NFκBIA*, etc.), apoptosis (DEGs: *FGF10*, *MAPK7*, etc.), etc. (Figure 13C). In the Vector and ADPN O/E groups, the DEGs were enriched in biological processes, such as the negative regulation of extrinsic apoptosis (DEGs: *FGF10*, *MAPK7*, *AGAP2*, *HMGB2*, etc.) and immune response (DEGs: *GPX2*, *MAPK7*, etc.), the regulation of hormone secretion (DEGs: *GLUL*, *CASR*, etc.), organ growth (DEGs: *TENM4*, *GGCX*, etc.) and cell junction maintenance (DEGs: *TENM4*, *F2R*, etc.) (Figure 13D). In the H_2_O_2_ and siADPN + H_2_O_2_ groups, the DEGs were enriched in the negative regulation of the cell cycle (DEGs: *HUS1*, *TTK*, etc.), the reproductive process (DEGs: *CALR*, *HUS1*, etc.), cell proliferation (DEGs: *TGFβ3*, *MEF2C*, etc.) and the immune effector process (DEGs: *TGFβ3*, *DHX58*, etc.) (Figure 13E). The ADPN O/E and ADPN O/E + H_2_O_2_ groups showed enriched biological processes, such as the regulation of the cell cycle (DEGs: *FGF10*, *PLK2*, etc.), hormone secretion (DEGs: *AGTR1*, *NRG1*, etc.), response to cAMP (DEGs: *COL1A1*, *AKAP6*, *PAX4*, etc.) and developmental induction (DEGs: *FGF10*, *GDNF*, etc.), the negative regulation of apoptosis (DEGs: *FGF10*, *NRG1*, *GDNF*, etc.), the positive regulation of DNA repair (DEGs: *FGF10*, etc.), etc. (Figure 13F).

The KEGG pathway database was used to conduct pathway analysis and identify enriched pathways in DEGs. The pathway enrichment analysis, results are shown in Figure 14. KEGG analysis of DEGs in the control and H_2_O_2_ groups showed significantly enriched pathways such as GnRH (DEGs: *KRAS*, *ITPR3*, etc.), VEGF (DEGs: *KRAS*, *PLA2G4F*, etc.), ECM-receptor interaction (DEGs: *SV2C*, *COL2A1*, etc.), ErbB (DEGs: *ERBB2*, *CDKN1A*, etc.) and apoptosis (DEGs: *MCL1*, *KRAS*, etc.) (Figure 14A). The NC and siADPN groups showed an enriched cell cycle (DEGc: *PCNA*, *TGFβ3*, *CDKN2B*, etc.), AGE-RAGE (DEGs: *AGTR1*, *STAT2*, etc.), PPAR (DEGs: *ME1*, *CPT1B*, etc.) and TGF-β (DEGs: *TGFβ3*, *SP1*, etc.) (Figure 14B). In the vector and ADPN O/E groups, the citrate cycle (DEGs: *BCKDHA*, etc.), arginine biosynthesis (DEGs: *GLUL*, *ARG2*, etc.), pyruvate metabolism (DEGs: *BCKDHA*, etc.) and ECM-receptor interaction (DEGs *COL1A1*, *COL1A2*, etc.) pathways were enriched (Figure 14C). In the H_2_O_2_ and siADPN + H_2_O_2_ groups, pathways were enriched for nitrogen metabolism (DEGs: *GLUL*, *CA8*) and AGE-RAGE (DEGs: *TGFβ3*, *COL1A1*, *AGTR1*, etc.) (Figure 14D), and in the H_2_O_2_ and ADPN O/E + H_2_O_2_ groups, AGE-RAGE (DEGs: *TGFβ3*, *COL1A1*, *AGTR1*, etc.) and TGF-β (DEGs: *TGFβ3*, etc.) pathways were enriched (Figure 14E).

## 3. Discussion

Unlike those in mammals, avian follicles ovulate in a strict hierarchical sequence and only a small number of primordial follicles develop into SYFs and progress to hierarchical follicles. The majority of follicles degenerate and undergo atresia before developing into SYFs or during the hierarchical development process, resulting in only a few follicles maturing and ovulating [1,2,5]. In geese with low egg production, follicular development and atresia have a direct impact on their poor breeding performance. Therefore, understanding the mechanism and regulation of follicular atresia in geese is crucial for improving geese productivity.

In the present study, changes were observed in atretic follicles, including alterations in their color, shape and elasticity, as well as the detachment of the granulosa cell layer. Notably, the blood vessels on the surface of the atretic follicles’ membrane disappeared. This phenomenon may be considered normal. According to a previous study, follicle-related blood vessels are temporary structures that are completely eliminated after ovulation or atresia [25].

Various factors in farming can induce oxidative stress, resulting in the generation of large amounts of ROS. Mitochondria-derived ROS are the primary inducers of apoptosis and autophagy during oxidative stress, mediating multiple signaling pathways; excessive accumulation of ROS in the follicular fluid can cause destruction and damage to proteins, enzymes, lipids and DNA. This can lead to apoptosis and autophagy in granulosa cells, ultimately resulting in follicular atresia [11,12,13]. Our findings are consistent with this hypothesis, as we observed high levels of oxidative stress, apoptosis and autophagy in atretic follicles of geese. It was also found that there was a high degree of oxidative stress in pre-hierarchical follicles compared to hierarchical follicles. This may be due to the follicle’s weak antioxidant capacity during the early stage of its development. However, it should be noted that autophagy was similarly enhanced in SYFs, which may explain why SYFs are susceptible to oxidative stress but have reduced expression of the pro-apoptotic protein Bax and elevated expression of the anti-apoptotic protein Bcl-2.

Autophagy can have both positive and negative effects. In normal conditions, autophagy can inhibit apoptosis, but excessive autophagy can also lead to apoptosis [26]. When follicles are stimulated to undergo oxidative stress, autophagy is activated to inhibit apoptosis. However, with continued oxidative stress, autophagy is gradually enhanced, ultimately resulting in follicular atresia. Furthermore, elevated levels of apoptosis were detected in F1 follicles, indicating their potential progression towards follicular atresia. As previously stated, poultry ovulation follows a strict hierarchical system. Therefore, if atresia occurs in an F1 follicle, a different F2 follicle needs to develop into an F1 follicle for ovulation to occur, resulting in longer intervals between laying and reduced egg production within the limited laying cycle.

GCs are crucial in follicular development, and their apoptosis leads to follicular atresia, which, in turn, affects reproductive function. Numerous studies have demonstrated that follicular atresia is caused by the apoptosis of follicular granulosa cells. During follicular atresia, the first observable change is nuclear shrinkage in the granulosa cells. This is followed by detachment of the granulosa cell layer, rupture of the basement membrane and, ultimately, hypertrophy of the membrane cells and disruption of the membrane vasculature. Additionally, granulosa cell apoptosis may occur before the morphological changes of follicular atresia appear, and follicular atresia can only be observed when granulosa cell apoptosis has reached a certain level [27,28,29].

By culturing primary GCs in vitro and constructing an oxidative stress model with the addition of H_2_O_2_, we demonstrated that oxidative stress results in the accumulation of ROS in goose GCs, reducing their cell viability and leading to apoptosis and autophagy. This is similar to the results of previous studies in chickens and pigs [30,31]. Notably, autophagy was enhanced in H_2_O_2_-treated GCs, and *P62* mRNA expression was also increased. Increased mRNA and protein expression of P62 was similarly found in atretic follicle tissues. This could be related to the blockage of autophagic flux.

Autophagy is a dynamic process that involves three sequential steps: autophagosome formation, autophagosome-lysosome fusion and degradation [32]. P62 carries ubiquitinated substrates that bind to LC3 and are then transported by autophagosomes to lysosomes for degradation [33]. Therefore, the level of P62 can be used as a marker to assess autophagic flux.

As our results demonstrate, P62 expression was increased in atretic follicles and H_2_O_2_-treated GCs, but the expression of autophagy marker molecules such as LC3, Beclin1, etc., was also found to be increased. This indicates that the autophagy process activated by oxidative stress in GCs may not be complete. When GCs were stimulated with H_2_O_2_, autophagy was initiated, but the degradation stage was blocked, leading to the inability to degrade the carried substrate despite the formation of an autophagosome. This resulted in the accumulation of P62 and substrate, ultimately leading to GC apoptosis. Recent research has demonstrated that blocked autophagic flux has a negative impact on the growth and differentiation of GCs [34].

ADPN has been extensively studied in its role in the regulation of glycolipid metabolism [35,36]. In recent years, there has been an increasing interest in the role of ADPN in regulating the reproductive system. Previous studies showed ADPN can stimulate spermatogenesis and protects germ cells from apoptosis and promote oogenesis and ovarian steroid hormones production [37,38]. Our study found that ADPN expression and antioxidant capacity was lower in atretic and pre-hierarchical follicles compared to hierarchical follicles, suggesting a correlation between ADPN and follicular antioxidant capacity. Luti et al. [39] similarly found a significant correlation between the total ADPN levels and the degree of oxidative stress in human follicular fluid and serum. In addition, Choubey et al. [38] demonstrated that ADPN enhances the antioxidant capacity of the reproductive system. In contrast, decreased synthesis of ADPN leads to increased oxidative stress and inhibition of testicular function [40]. It has been previously confirmed that ADPN is expressed in goose ovaries and regulates steroid hormone secretion in GCs [24]. In chickens, the activation of ADPN was found to promote progesterone secretion and inhibit estrogen secretion in GCs, which further affects ovarian function [20]. ADPN also indirectly affects reproductive function in pigs by influencing the gonadal secretion of LH and FSH [41]. In addition, ADPN promotes the proliferation of GCs and regulates follicular growth and embryonic development in bovines, mice, and humans [18,42].

Although ADPN has been reported to regulate apoptosis and autophagy [20,21], its role in the regulation of oxidative stress and death in geese GCs is not well understood. In this study, ADPN was found to regulate oxidative stress and apoptosis in GCs. Overexpression of ADPN reduced ROS accumulation and increased resistance to apoptosis. Conversely, knockdown of ADPN expression had the opposite effect. It is this beneficial effect of ADPN that allows it to mitigate the effects of H_2_O_2_ on GCs. Knockdown of ADPN exacerbated the degree of oxidative stress and apoptosis in GCs caused by H_2_O_2_, while overexpression of ADPN alleviated these effects. These effects of ADPN may be mediated by binding to AdipoR1, as we detected an increase or decrease in the expression levels of AdipoR1 upon overexpression of ADPN and knockdown of ADPN.

Meanwhile, it has been found in both our research and the research of others [43,44,45] that ADPN can activate autophagy. Overexpression of ADPN can activate the autophagy of GCs and the knockdown of ADPN expression can inhibit it. In the model of oxidative stress in GCs, the knockdown of ADPN still inhibited autophagy, but overexpression of ADPN did not activate autophagy, and it reduced the expression of both LC3 and Beclin1 in GCs. This suggested that ADPN may play a role in the restoration of obstructed autophagic flux. Previous studies have reported the role of ADPN in regulating autophagic flux. For instance, ADPN activates autophagic flux to reduce endoplasmic reticulum stress and insulin resistance in skeletal muscle cells [46]. ADPN reduces insulin resistance induced by a high-fat diet and decreases oxidative stress by promoting autophagic flux in skeletal muscle cells [43]. Furthermore, ADPN was observed to restore the autophagic process and activate autophagic flux to protect hepatocytes from ethanol-induced apoptosis by regulating ATG5 expression through the AMPK/FoxO3A axis [47].

GO analysis showed that, in addition to inducing apoptosis in GCs, oxidative stress affects the cell cycle and differentiation of GCs, promoting inflammation and suppressing the immune response. GO analyses in the siRNA group were similar to those in the H_2_O_2_ group. Conversely, overexpression of ADPN lead to the enrichment of biological processes that benefit the growth of GCs, such as anti-apoptosis, regulation of hormone secretion, organ growth and maintenance of cell junctions. KEGG analysis showed that pathways, such as metabolism, PPAR, GnRH, VEGF, ECM-receptor interaction, ErbB, apoptosis, cell cycle, TGF-β and AGE-RAGE, etc., were enriched in response to overexpression of ADPN.

As previously mentioned, ADPN regulates glycolipid metabolism and modulates the secretion of reproductive hormones. Studies have reported that ADPN binds to its receptor and promotes fatty acid oxidation through the AMPK-PPAR pathway [48]. ADPN can regulate ovarian activities through autocrine/paracrine effects, and previous studies showed that ADPN modulates ovarian steroid hormone secretion in a variety of organisms (e.g., bovines, mice, pigs, etc.) [49]. In addition, the ADPN receptor agonist AdipoRon promotes the secretion of reproductive hormones and gonadal development through the hypothalamic-pituitary-gonadal (HPG) axis, and GnRH is an important regulator of the HPG axis [50]. Our aforementioned results suggest that oxidative stress and ADPN have a function in regulating apoptosis. Taken together, our unpublished results and other studies suggest that ADPN has a role in resisting cellular senescence [51,52]. Therefore, it is not surprising that metabolism, PPAR, GnRH, apoptosis and cell cycle pathways were enriched in response to its overexpression.

ErbB is a member of the EGFR family and it has been claimed that the EGF/EGFR pathway regulates GC and follicle development [53]. In addition, inhibition of ErbB induced apoptosis and oxidative stress, whereas overexpression of ErbB increased the expression of antioxidant enzymes and decreased ROS levels [54,55]. The ECM is involved in the regulation of ovarian physiological activities, and its interactions with the receptor are critical for ovarian development [56,57]. VEGF is closely associated with angiogenesis, and VGEF has also been found to regulate the activity and apoptosis of GCs in yak [58]. Luo et al. [59] also demonstrated that upregulation of VEGF expression resists apoptosis and oxidative stress through the Notch 1-Jagged 1 signaling pathway.

Based on the enriched GO terms and KEGG pathways, we focused on the TGF-β and AGE-RAGE pathways. The GO analysis in this study showed that *FGF10* was involved in the regulation of apoptosis, cell cycle, DNA repair and other biological processes. FGF10 may increase cellular sensitivity by upregulating TGF-β expression and augmenting its regulatory effects on cells [60]. TGFβ was able to modulate oxidative stress. For example, knockdown of TGF-β1 and TGF-β2 increased the accumulation of ROS in retinal ganglion cells, and TGFβ1 also protected chondrocytes from oxidative stress [61,62]. Some previous studies also showed that TGF-β can regulate autophagy and cell apoptosis [63,64,65]. Furthermore, TGF-β can interact with EGFR and VEGF to regulate apoptosis. For instance, TGF-β can resist apoptosis by increasing the expression of EGFR ligands and reducing inflammation and promoting VEGF-dependent angiogenesis [66,67]. The AGE-RAGE pathway was reported to induce oxidative stress and apoptosis in a variety of cells [68,69]. AGE-RAGE can interact with TGF-β to regulate apoptosis, and RAGE deletion in RAGE knockout mice affected TGF-β responsiveness and enhanced anti-apoptotic capacity in kidneys [70]. The RNA-seq results demonstrate the role of oxidative stress and ADPN in regulating GCs. However, in this study, the limited laying cycle of geese and the existence of individual differences made it challenging to obtain a sufficient number of primary cell samples that could be used to fulfil the experimental requirements. Therefore, it is unclear how oxidative stress leads to the blockage of autophagic flux in GCs and the molecular mechanism by which ADPN regulates this process. These two scientific issues will be further explored in our future studies.

## 4. Materials and Methods

### 4.1. Ethics Statement

All experimental procedures in this study were approved by the animal welfare committee of the College of Animal Science and Veterinary Medicine of Shenyang Agricultural University (No. 202006006).

### 4.2. Animals and Sample Collection

Sixty 10-month-old laying geese were obtained from the Liaoyang Huoyan Goose Stock Breeding Farm in Liaoyang City, China. After anesthesia and sacrifice of the geese by exsanguination through the jugular vein, the ovaries were dissected using sterilized scissors and forceps. The ovaries were then washed in a Petri dish filled with normal saline on ice to distinguish between healthy and atretic follicles based on their external morphological characteristics, such as shape, color, elasticity and the presence of hemorrhagic spots on their surface. The follicles were classified into categories based on their size and stored at −80 °C for future use. Additionally, the follicles from each period were fixed in a 4% paraformaldehyde and glutaraldehyde fixative for subsequent Hematoxylin and Eosin (H and E) staining and transmission electron microscopy observation.

### 4.3. Hematoxylin and Eosin (H and E) Staining

The follicles were fixed in 4% paraformaldehyde for 72 h, rinsed under running water for 24 h, dehydrated using gradient ethanol, transferred to gradient xylene and embedded in paraffin. Sections of approximately 4 μm were cut, stained with hematoxylin and eosin and observed under a light microscope (Nikon Eclipse Ti-SR, Tokyo, Japan).

### 4.4. Measurement of SOD and MDA Content in Follicular Tissue

Commercially available kits were used to measure SOD (Elabscience, Wuhan, China, E-BC-K020-M) and MDA (Nanjing Jiancheng Bioengineering Institute, Nanjing, China, A003-1-2) levels in follicular tissue, following the manufacturer’s instructions.

### 4.5. Quantitative Real-Time PCR

Total RNA was extracted from follicular tissue and GC samples using TRIzol reagent (Vazyme, Nanjing, China, R411-01) according to the manufacturer’s instructions. RNA purity and concentration were detected by measuring the absorbance ratio at 260 nm and 280 nm (NanoDrop, Thermo Scientific, Waltham, MA, USA). The cDNA was synthesized according to the manufacturer’s instructions using the HiScript III 1st Strand cDNA Synthesis Kit with gDNA wiper (Vazyme, Nanjing, China, R312-02). The mRNA expression levels of related genes were quantified using Taq Pro Universal SYBR qPCR Master Mix (Vazyme, Nanjing, China, Q712-02). Each 20 μL reaction volume contained 10 μL 2 × Taq Pro Universal SYBR qPCR Master Mix, 1 μL cDNA template, 0.4 μL forward and reverse primer sand 8.2 μL ddH_2_O. The reaction procedure included pre-denaturation at 95 °C for 30 s, 40 cycles of denaturation at 94 °C for 10 s and specific annealing and extension at 60 °C for 30 s. The primers used for the RT-qPCR are listed in Table 1. The relative expression of the targeted gene was calculated using the 2^−ΔΔCt^ method, and beta-actin was designated as an internal reference gene.

### 4.6. Western Blot

The follicular tissue was lysed on ice with RIPA Lysis Buffer (Beyotime, Shanghai, China, P0013B) and supplemented with a 1% protease inhibitor cocktail (Beyotime, Shanghai, China, P1005). Protein concentrations were determined using the BCA Protein Assay Kit (Beyotime Shanghai, China, P0012). Samples containing 50 μg protein were separated on 10–15% SDS-PAGE gels and subsequently electro-transferred onto a polyvinylidene difluoride (PVDF) membranes (Millipore, Darmstadt, France). The membranes were blocked with 5% non-fat milk for 1 h at room temperature. Subsequently, they were incubated overnight at 4 °C with anti-β-actin (1:8000, Immunoway, Plano, TX, USA, YT0099), anti-GAPDH (1:8000, Immunoway, Plano, TX, USA, YN5585), anti-Bax (1:1000, Proteintech, Chicago, IL, USA, 50599-2-Ig), anti-Bcl-2 (1:1000, Immunoway, Plano, TX, USA, YT0470), anti-ADPN (1:1000, Proteintech, Chicago, IL, USA, YT0091), anti-AdipoR1 (1:1500, Abmart, Shanghai, China, PK14733), anti-AdipoR2 (1:1000, Affintiy, OH, USA, DF12811), anti-microtubule-associated protein light chain 3B (LC3B) (1:2000, Abcam, Cambridge, UK, ab192890) or anti-Sequestosome 1 (p62/SQSTM1) (1:1000, Proteintech, Chicago, IL, USA, YT7058 ) primary antibodies. The membranes were washed three times with TBST for 10 min and then were incubated with a secondary horseradish peroxidase-conjugated antibody (1:8000, Proteintech, Chicago, IL, USA, PR30009) at 37 °C for 1 h. The membranes were visualized using a SuperPico ECL Chemiluminescence Kit (Vazyme, Nanjing, China, E422-01). The stained proteins were quantified densitometrically using ImageJ (v1.5.3s) software (National Institutes of Health, Bethesda, MD, USA).

### 4.7. Transmission Electron Microscopy (TEM)

The follicles were fixed in 2.5% glutaraldehyde for TEM and dehydrated in ethyl alcohol and acetone. They were then embedded in epoxy resin. Ultrathin sections, with a thickness of 50–70 nm, were cut using a diamond knife and double-stained with uranyl acetate and lead citrate. The ultra-microstructure of the samples was examined using a transmission electron microscope (HITACHI, Tokyo, Japan).

### 4.8. Cell Culture Identification and Transfection

Hierarchical follicles were collected, and GCs were isolated according to the method of Gilbert et al. [71], and digested and cultured according to the method of Kang et al. [72]. The GCs were cultured in an incubator at 38.5 °C with 5% CO_2_ until the cell fusion rate exceeded 80%. The M199 (Gibco, Carlsbad, CA, USA, 11150059) complete medium (with 15% FBS) was changed every 24 h. The identification of GCs was based on the expression of FSHR (Bioss, Beijing, China, bs-0895R), which is a specific marker for GCs. siRNA-ADPN sequences (Table 2) and pIRES2-EGFP-ADPN plasmids were designed and synthesized based on the goose ADPN gene sequence (NCBI Gene ID: 106040284). The pIRES2-EGFP plasmids were purchased from Bio-Transduction Lab Co., Ltd. (Wuhan, China) siRNA-NC, and siRNA-ADPN were synthesized by GenePharma (Suzhou, China). GCs were seeded into six-well plates and cultured for 24 h until 60% confluence. Cell transfection was performed according to Lipofectamine 3000 (Invitrogen, Carlsbad, CA, USA, L3000015) instructions.

### 4.9. Cell Viability Assay

GCs were seeded in 96-well plates until the cells grew to 90% confluency. Then, Cell Counting Kit-8 (CCK-8; Elabscience, Wuhan, China, E-CK-A362) was used to measure the cell viability following the manufacturer’s instructions.

### 4.10. ROS Staining Assay

ROS accumulation was measured using the fluorescent ROS probe dihydroethidium (DHE) (GLPBIO, Montclair, CA, USA, GD20803). After treatment, cells were washed three times with PBS, and 10 μmol/L DHE in non-phenol red medium was added to the wells. The cells were then incubated at 37 °C for 30 min, protected from light, and washed three times with PBS. Finally, they were observed under a fluorescence microscope (Leica DMI8, Wetzlar, Germany).

### 4.11. Immunofluorescence

Cells were plated on coverslips, fixed with 4% paraformaldehyde for 1 h, washed three times with PBS and permeabilized with 0.5% Triton X-100 for 20 min. They were then treated in PBS with 2% bovine serum albumin (BSA) for 1 h and incubated overnight at 4 °C with anti-Caspase-3 antibody (1:100; Proteintech, Chicago, IL, USA, 66470-1-lg) or anti-Beclin1 antibody (1:100; Proteintech, Chicago, IL, USA, 11306-1-ap) diluted in 2% BSA. Finally, the cells were washed with PBS and incubated with a FITC-conjugated secondary antibody for 1 h in the dark. After washing, the cells were stained in PBS with DAPI. Subsequently, they were observed under a fluorescent microscope (Leica DMI8, Wetzlar, Germany).

### 4.12. Flow Cytometry

GCs were digested with 0.25% trypsin and collected to produce a cell suspension, which was then washed three times with PBS. Flow cytometry was used to analyze apoptosis in the GCs using an Annexin V-Elab Fluor^®^647/PI Apoptosis Kit (Elabscience, Wuhan, China, E-CK-A213) according to the manufacturer’s instructions.

### 4.13. Enzyme-Linked Immunosorbent Assay (ELISA)

Commercially available ADPN ELISA kits were used to measure ADPN (Nanjing Jiancheng Bioengineering Institute, Nanjing, China, H179-1-2) levels in GCs and the culture supernatant of GCs, following the manufacturer’s instructions.

### 4.14. RNA Sequencing and Data Processing

The RNA sequencing was conducted by the APExBIO Chinese distributor (Shanghai, China). Briefly, the RNA of GCs was extracted and tested for concentration and purity. Subsequently, double-stranded cDNA was synthesized and the product was purified. The library was then constructed and sequenced after quality control. Following sequencing, quality assessment of the fastq files using FastQC (0.12.1) software was carried out to filter out low-quality data. HISAT2 (v2.0.4) software was used to compare the clean data with the goose reference genome sequence (GCF_002166845.1) to obtain the comparison reference genome. Transcript reconstruction was performed using StringTie (2.1.7) software, assembled to obtain accurate transcript results and the expression of each gene or transcript was counted. Differential expression analysis was performed using DESeq2 (R4.1.2), and finally, the relevant pathways were found via enrichment analysis (KEGG, GO). Genes with *p* ≤ 0.05 and |Fold Change| ≥ 1.5 were classified as differentially expressed genes.

### 4.15. Statistical Analysis

Data are presented as the mean ± SEM unless indicated otherwise. Data were analyzed using a *t*-test and analysis of variance (ANOVA) with GraphPad Prism version 5.0 (GraphPad Software, San Diego, CA, USA). *p* <0.05 was considered to indicate statistical significance.

## 5. Conclusions

The accumulation of ROS caused by oxidative stress can lead to apoptosis and the blockage of autophagic flux in GCs, resulting in follicular atresia. However, adiponectin (ADPN) protects GCs from oxidative stress and has anti-apoptotic and autophagy-regulating effects.

## Figures and Tables

**Figure 1 ijms-25-05400-f001:**
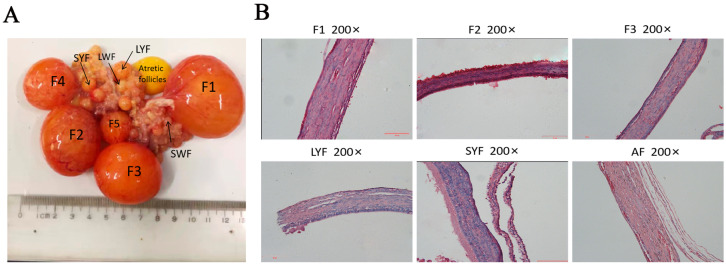
Morphological and histological characteristics of follicles at various stages in geese. (**A**) Follicles at different stages of development and atretic follicles. Hierarchical follicles are arranged in ascending order of size. Follicles 1–2 mm in diameter are SWFs, those 2–5 mm are LWFs, those 5–10 mm are SYFs, and pre-hierarchical follicles > 10 mm in diameter are LYFs. (**B**) H and E staining showing histological features of each membrane layer of the follicle. AF represents an atretic follicle. Scale bar = 50 μm.

**Figure 2 ijms-25-05400-f002:**
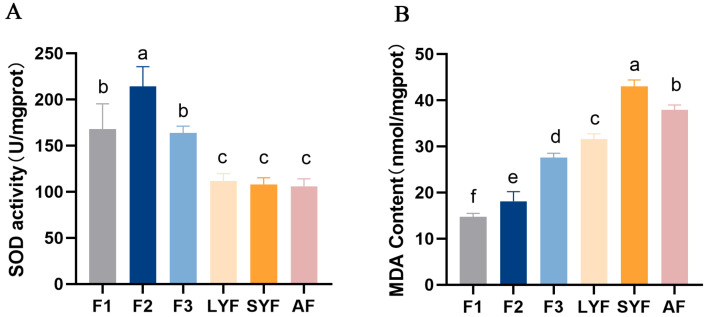
Oxidative stress levels were measured in various follicles. (**A**) SOD activity. (**B**) MDA content. The letters above each bar indicate the significance of the differences. Differences with the same letter were not considered significant, while those with different letters were considered significant (*p* < 0.05). The values in the bars with vertical lines represent the mean ± SEM, which are the same as in the following figures.

**Figure 3 ijms-25-05400-f003:**
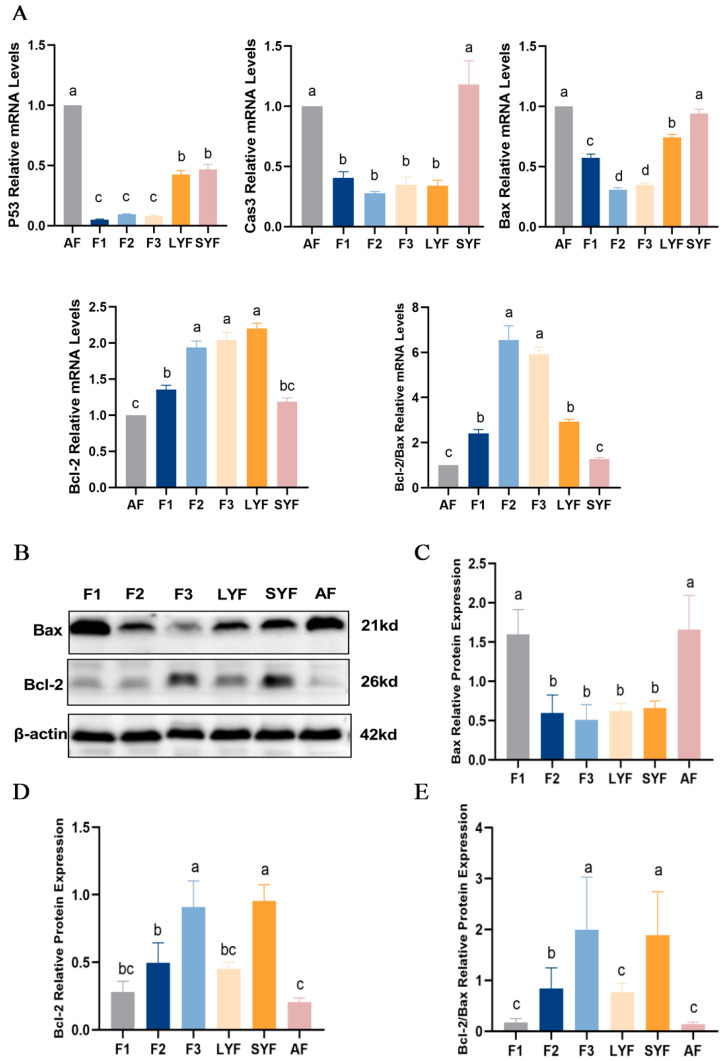
The level of apoptosis in goose follicles at different stages. (**A**) The mRNA expression of *P53*, *Caspase-3*, *Bax* and *Bcl-2*. (**B**–**D**) The protein expression of Bax and Bcl-2 with β-actin as an internal control. (**E**) The ratio of Bcl-2 to Bax protein expression. Different letters were considered significant (*p* < 0.05).

**Figure 4 ijms-25-05400-f004:**
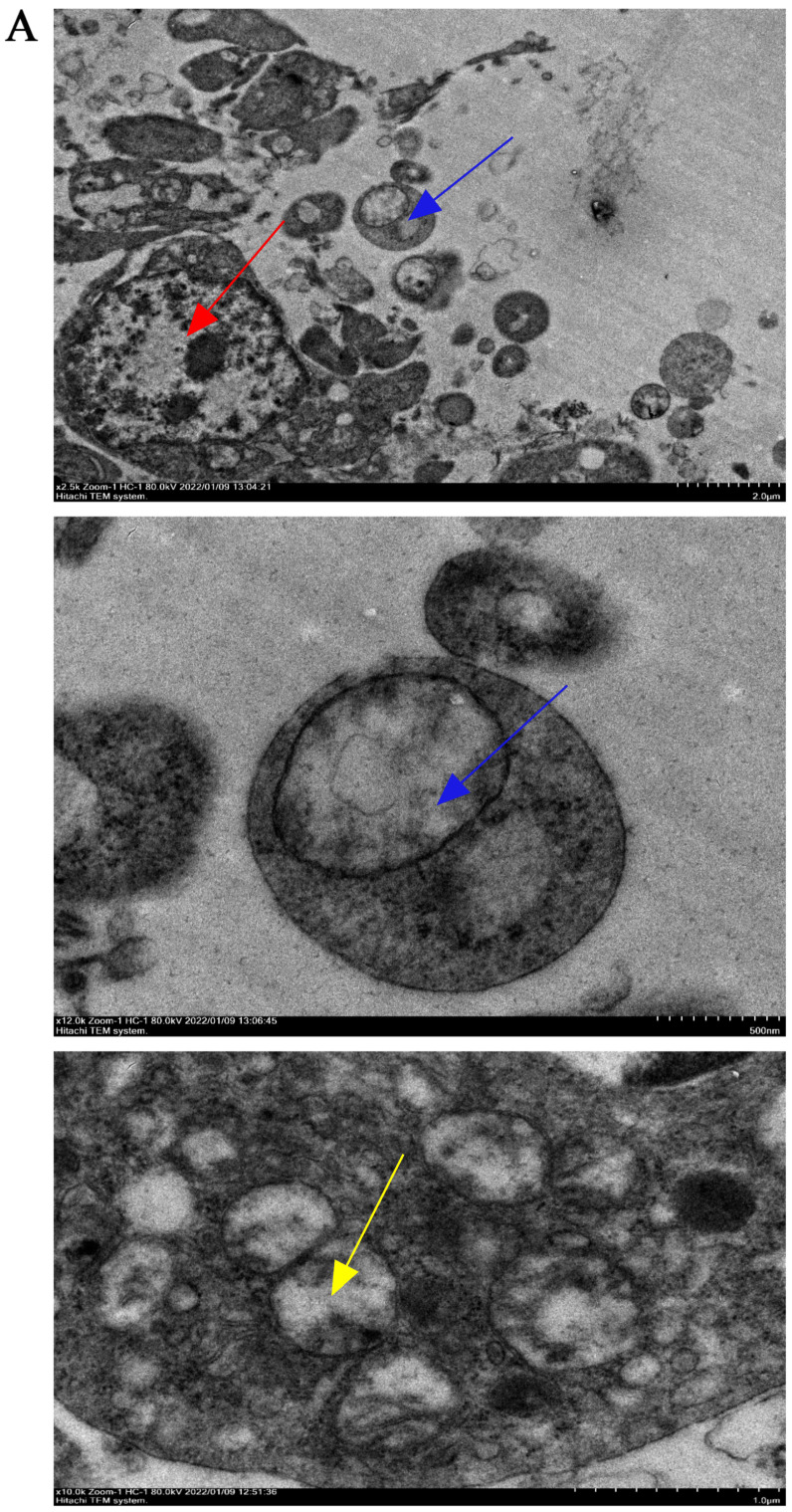
The level of autophagy in goose follicles at different stages. (**A**) The submicroscopic structure of an atretic follicle. The lysed nucleus is indicated by the red arrowheads, while the cristae-vanishing mitochondria encapsulated by autophagosomes are indicated by the blue arrowheads. The yellow arrowheads indicate cristae-vanishing mitochondria. The scale bars from left to right are 2.0 μm, 500 nm and 1.0 μm, respectively. (**B**) The mRNA expression of *ATG5*, *ATG7*, *Beclin 1*, *P62* and *LC3*. (**C**–**F**) The protein expression of P62, LC3-I and LC3-II normalized to that of β-actin. (**G**) The ratio of LC3-II to LC3-I protein expression. Different letters were considered significant (*p* < 0.05).

**Figure 5 ijms-25-05400-f005:**
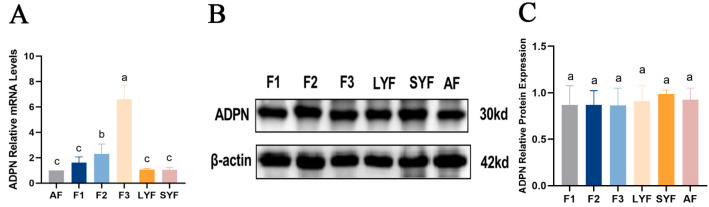
The expression of ADPN in goose follicles at different stages. (**A**) The mRNA expression of *ADPN*. (**B**,**C**) The protein expression of ADPN. Different letters were considered significant (*p* < 0.05).

**Figure 6 ijms-25-05400-f006:**
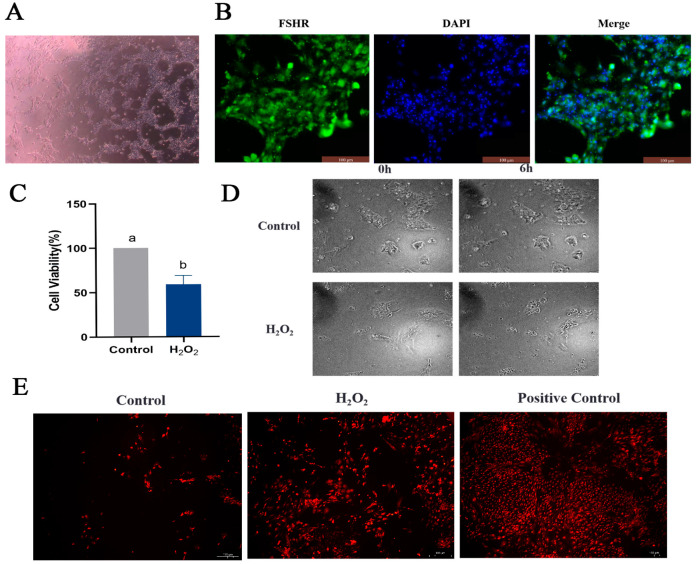
Cultivation and identification of GCs and establishment of an in vitro oxidative stress model. (**A**) GCs isolated and cultured from goose follicles; GC images are shown under 100× magnification. (**B**) The expression of the follicle stimulating hormone receptor (FSHR) in GCs determined via immunofluorescence; scale bar = 100 μm. (**C**) Viability of GCs as examined via CCK8 assay. (**D**) Observation of morphological changes in GCs using live cell imaging; scale bar = 200 μm. (**E**) ROS determination using the dihydroethidium (DHE) probe; scale bar = 100 μm. Different letters were considered significant (*p* < 0.05).

**Figure 7 ijms-25-05400-f007:**
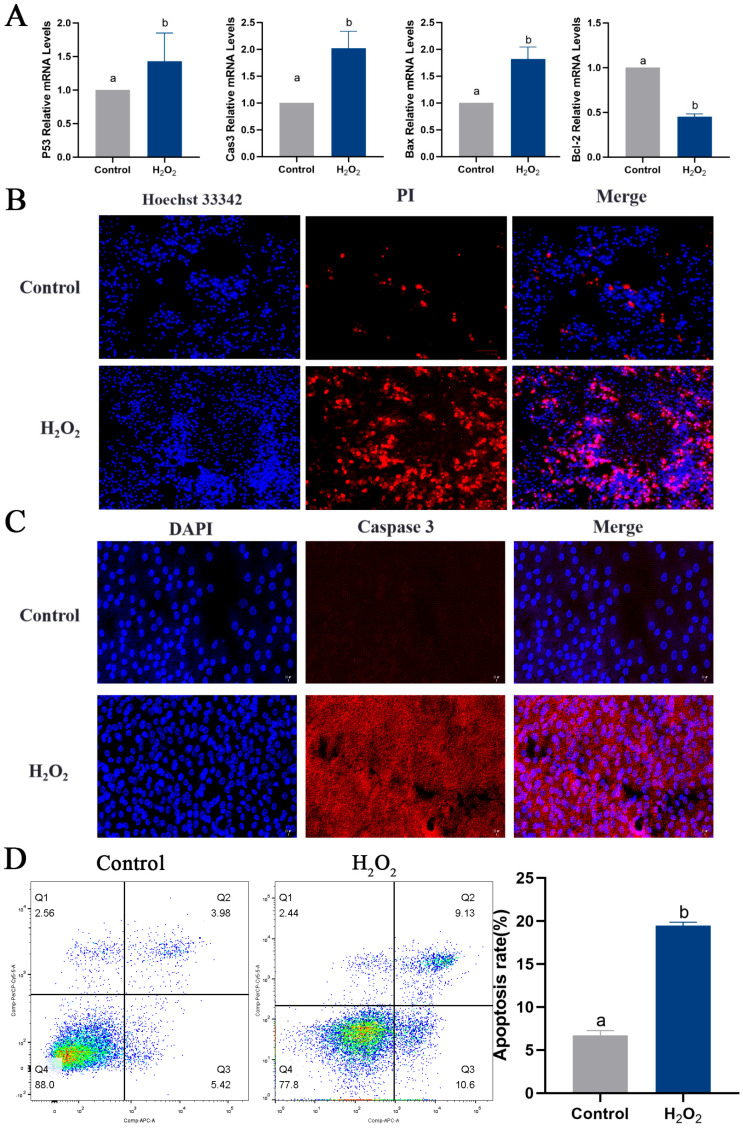
Oxidative stress conditions induce apoptosis and autophagy in GCs. (**A**) The expression levels of *P53*, *Caspase-3*, *Bax* and *Bcl-2* were measured. (**B**) Apoptosis in GCs determined via Hoechst33342/PI double staining; scale bar = 100 μm. (**C**) The expression of Caspase-3 protein detected using immunofluorescence; scale bar = 20 μm. (**D**) Apoptosis rate of GCs determined via flow cytometry. Apoptosis rate (%) = (number of cells in Quadrant 2 + Quadrant 3)/total number of cells × 100%. (**E**) The expression of *ATG5*, *ATG7*, *Beclin 1*, *P62* and *LC3*. (**F**) The expression of Beclin 1 protein detected using immunofluorescence; scale bar = 20 μm. Different letters were considered significant (*p* < 0.05).

**Figure 8 ijms-25-05400-f008:**
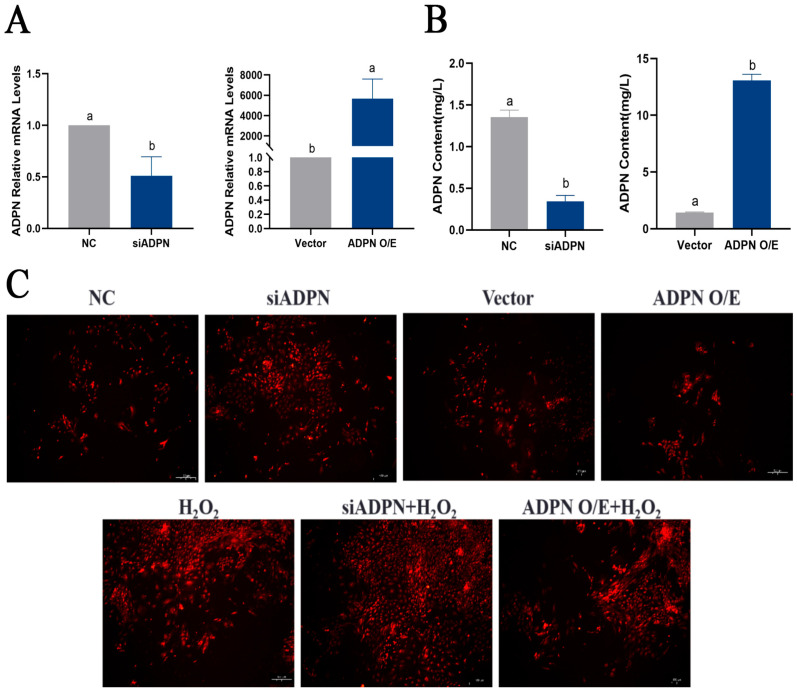
ADPN regulates the production of ROS and influences the generation of ROS induced by H_2_O_2_. (**A**) Expression of *ADPN* mRNA after siRNA-mediated ADPN knockdown and overexpression plasmid-mediated ADPN overexpression. (**B**) ADPN secretion in culture supernatants of GCs after ADPN knockdown and overexpression. (**C**) Effect of ADPN knockdown and overexpression on ROS production in GCs; scale bar = 100 μm. Different letters were considered significant (*p* < 0.05).

**Figure 9 ijms-25-05400-f009:**
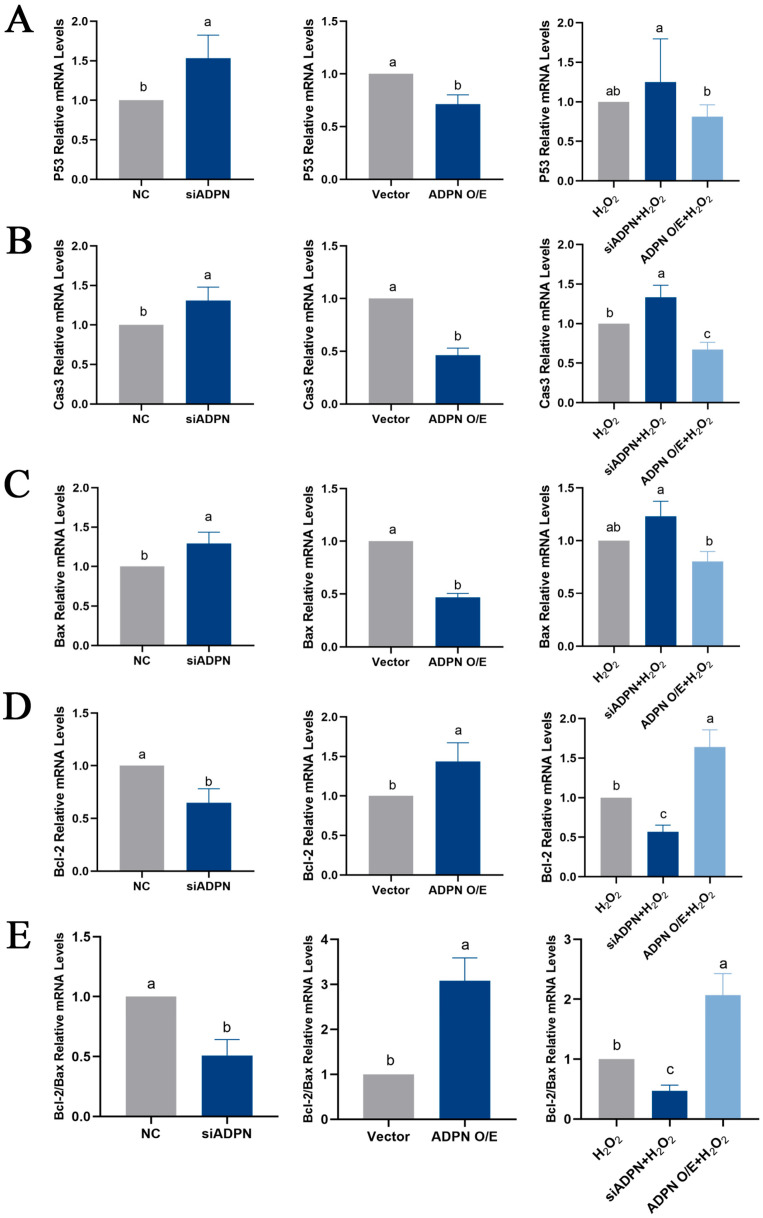
Effects of ADPN knockdown and overexpression on mRNA expression of apoptosis-related genes in GCs. (**A**) RT-qPCR analysis of *P53* gene expression. (**B**) RT-qPCR analysis of *Caspase-3* gene expression. (**C**) RT-qPCR analysis of *Bax* gene expression. (**D**) RT-qPCR analysis of *Bcl-2* gene expression. (**E**) The ratio of *Bcl-2* to *Bax* mRNA expression. Different letters were considered significant (*p* < 0.05).

**Figure 10 ijms-25-05400-f010:**
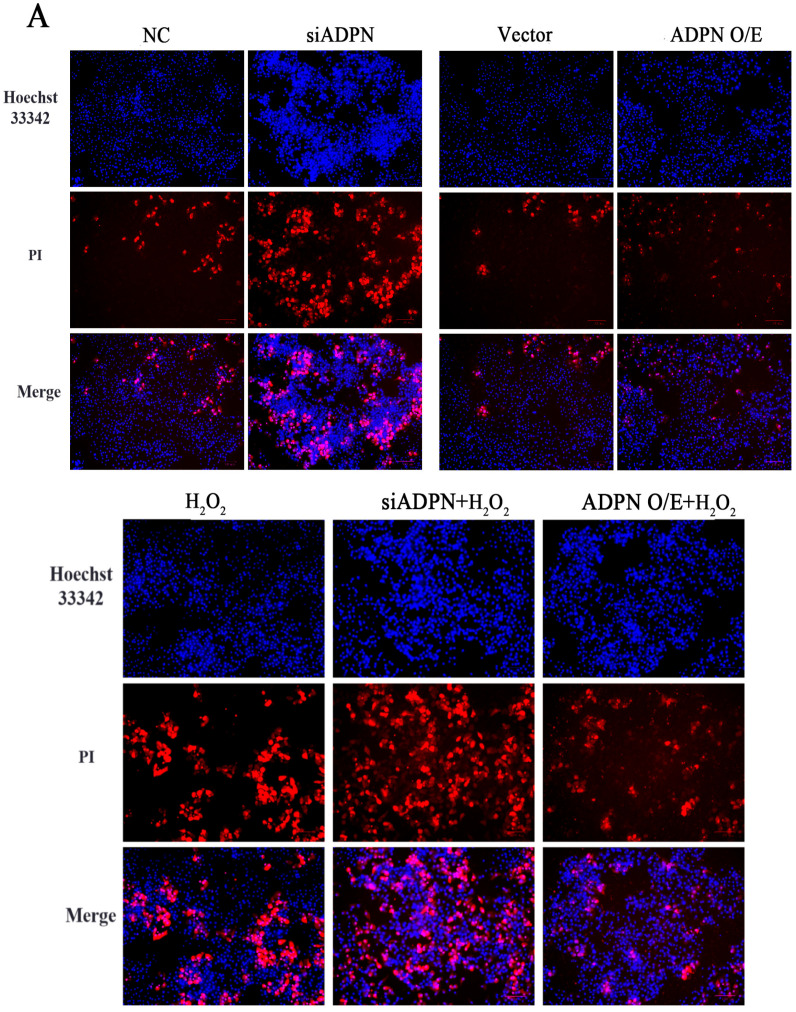
The role of ADPN knockdown and overexpression apoptosis of GCs. (**A**) Apoptosis in GCs determined using Hoechst33342/PI staining; scale bar = 100 μm. (**B**) The expression of Caspase-3 protein detected using immunofluorescence; scale bar = 20 μm. (**C**,**D**) Flow cytometry was used to track apoptosis and calculate the apoptosis rate. Different letters were considered significant (*p* < 0.05).

**Figure 11 ijms-25-05400-f011:**
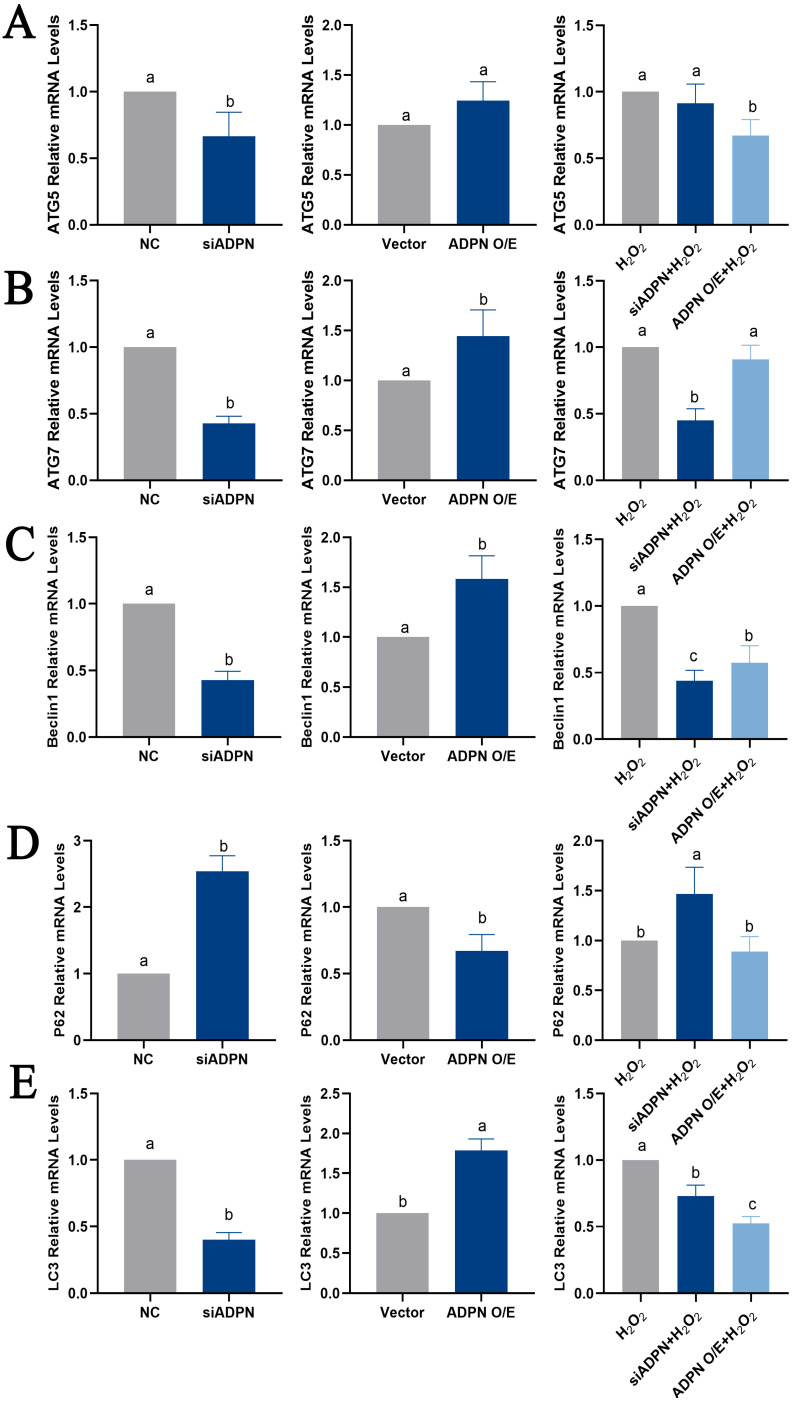
Effects of ADPN knockdown and overexpression on mRNA expression of autophagy-related genes in GCs. (**A**) RT-qPCR analysis of *ATG5* gene expression. (**B**) RT-qPCR analysis of *ATG7* gene expression. (**C**) RT-qPCR analysis of *Beclin 1* gene expression. (**D**) RT-qPCR analysis of *P62* gene expression. (**E**) RT-qPCR analysis of *LC3* gene expression. (**F**) The expression of Beclin 1 protein detected using immunofluorescence; scale bar = 20 μm. Different letters were considered significant (*p* < 0.05).

**Figure 12 ijms-25-05400-f012:**
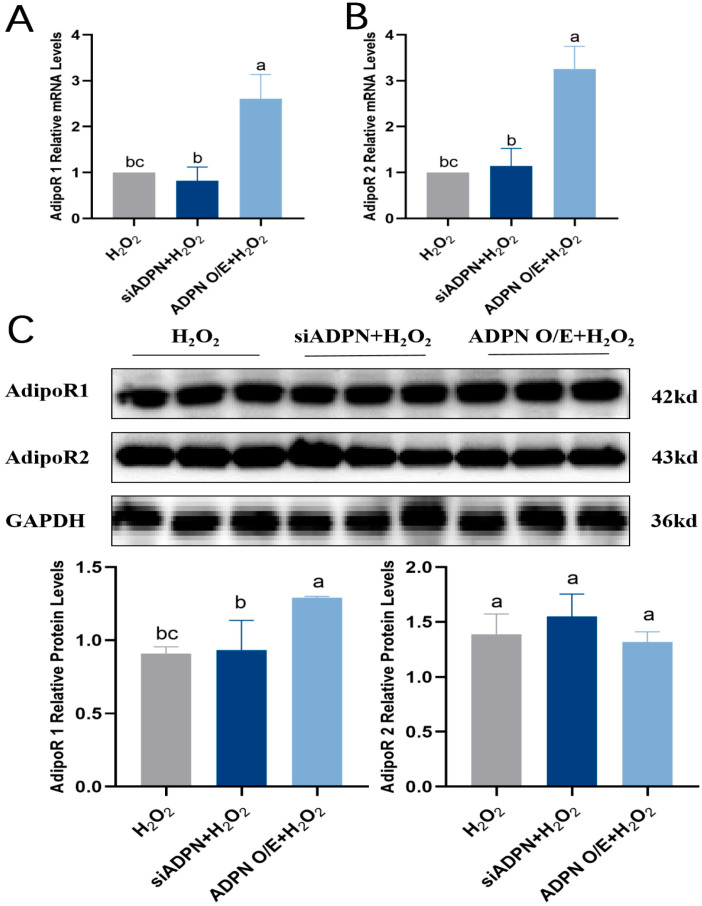
Effects of ADPN overexpression and knockdown on AdipoR1 and AdipoR2 expression levels in GCs. (**A**) RT-qPCR analysis of *AdipoR1* mRNA expression. (**B**) RT-qPCR analysis of *AdipoR2* mRNA expression. (**C**) The protein expression of AdipoR1 and AdipoR2. Different letters were considered significant (*p* < 0.05).

**Figure 13 ijms-25-05400-f013:**
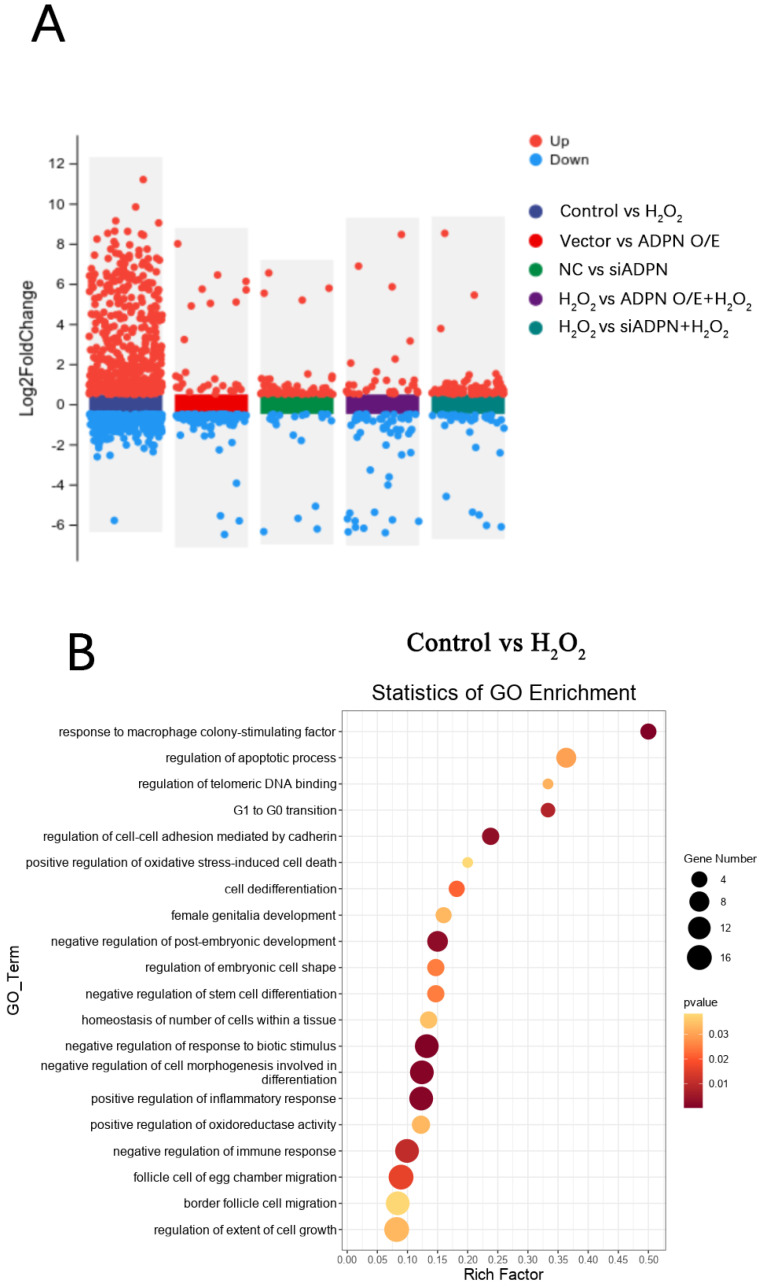
DEGs and GO enrichment in each group of GCs. (**A**) Volcano plot showing DEGs (*p* < 0.05, |FoldChange| ≥ 1.5). (**B**) GO function enrichment of DEGs between control and H_2_O_2_ groups. (**C**) GO function enrichment of DEGs between NC and siADPN groups. (**D**) GO function enrichment of DEGs between vector and ADPN O/E groups. (**E**) GO function enrichment of DEGs between H_2_O_2_ and siADPN + H_2_O_2_ groups. (**F**) GO function enrichment of DEGs between H_2_O_2_ and ADPN O/E + H_2_O_2_ groups.

**Figure 14 ijms-25-05400-f014:**
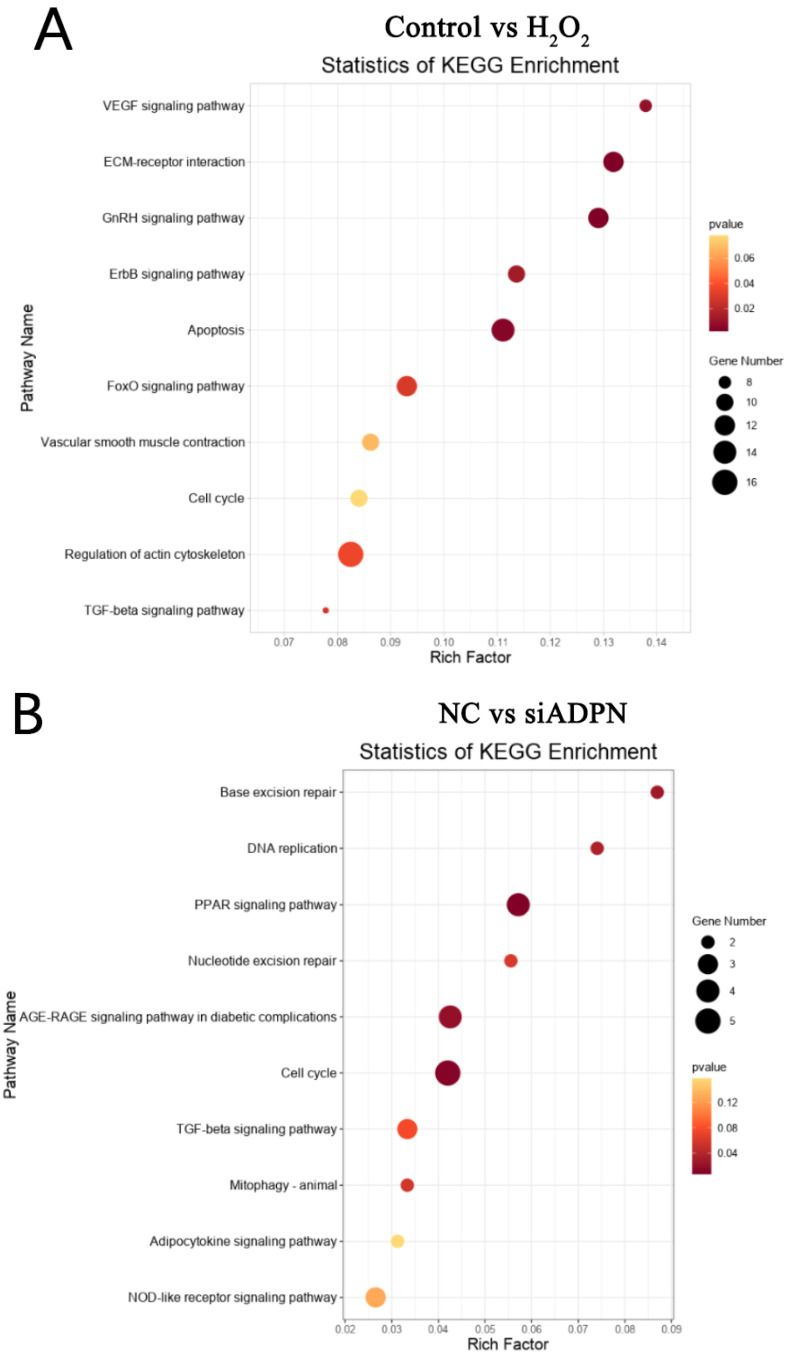
KEGG enrichment in each group of GCs. (**A**) KEGG pathways enrichment of DEGs between control and H_2_O_2_ groups. (**B**) KEGG pathway enrichment of DEGs between NC and siADPN groups. (**C**) KEGG pathway enrichment of DEGs between vector and ADPN O/E groups. (**D**) KEGG pathway enrichment of DEGs between H_2_O_2_ and siADPN + H_2_O_2_ groups. (**E**) KEGG pathway enrichment of DEGs between H_2_O_2_ and ADPN O/E + H_2_O_2_ groups.

**Table 1 ijms-25-05400-t001:** Primer sequences of quantitative real-time PCR.

Genes	Sequences (5′→3′)	Accession No.	Size (bp)
*ADPN*	FP: AACGAGCAGAACCACTAC RP: CGCCTTGTCCTTCTTGTA	KP993199	132
*AdipoR1*	FP: AAGTTGGATTATTCAGGAA RP: AATGGAGAGGTAGATGAG	KP993200	108
*AdipoR2*	FP: ATACTGAACAAGGCCACTATTT RP: CACCTGAATGCCTTACTCTC	KP993201	123
*p*53	FP: AGGAGGAGAACTTCCGCAAGAGG RP: CGTCGTTGATCTCCTTCAGCATCTC	XM_038171819.1	97
*Caspase*-3	FP: AGTGGACCAGATGAAATGAC RP: AGACTGAATAAACCAGGAGC	XM_013179825	126
*Bax*	FP: CTTCTCGGGTTTCTTGAGG RP: AACGCAGCAGGTGTAGGA	KY788660	200
*Bcl*-2	FP: TGACCGAGTACCTGAACCG RP: GCTCCCACCAGAACCAAA	XM_013187395	154
*ATG*5	FP: GATGAAATAACTGAAAGGGAAGC RP: TGAAGATCAAAGAGCAAACCAA	NM_001006409.1	124
*ATG*7	FP: CGACCAGTATGAACGAGA RP: CTGATGTAATAAAGTTAGACCC	XM_048046141	100
*Beclin*1	FP: GCACGCCCTCGCTAACA RP: GCAGTCCAAGAAAGCCACC	XM_048051864	184
*p*62	FP: GGTGGTGGGTGCTAGATTCAAGTG RP: TGTGCTCCTTGTGGATGCCTTTAC	XM_048057319.1	89
*LC3B*	FP: TGCTAACCAAGCCTTCTTCCTC RP: TCCTGCGAGGCATAAACCAT	NM_001364358	129
*β-actin*	FP: ATTGTCCACCGCAAATGCTTC RP: AAATAAAGCCATGCCAATCTCGTC	M26111	113

**Table 2 ijms-25-05400-t002:** Primer sequences for quantitative real-time PCR.

Name	Sequences (5′-3′)
siRNA-*ADPN*	FP: GGGACAACAACGGUGUCUATT RP: UAGACACCGUUGUUGUCCCTT
siRNA-NC	FP: UUCUCCGAACGUGUCACGUTT RP: ACGUGACACGUUCGGAGAATT

## Data Availability

The data used to support the findings of this study are available upon request from the corresponding author.

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
