# Peer review of "ADPN Regulates Oxidative Stress-Induced Follicular Atresia in Geese by Modulating Granulosa Cell Apoptosis and Autophagy"

_ijms, 2024, doi:10.3390/ijms25105400_

Round 1
Reviewer 1 Report
Comments and Suggestions for Authors
Manuscript ijms-2986959, entitled “ADPN Regulates Oxidative Stress-induced Follicular Atresia in Geese by Modulating Granulosa Cells Apoptosis and Autophagy”
The present article provides useful information about the role of adiponectin in oxidative stress-induced follicular atresia in geese. It is in general appropriately organized, carried out and written, however there are some minor points that should be corrected or clarified.
Title: Please use “Adiponectin” instead of “ADPN”
L60-61: “…anti-apoptotic properties. ADPN can reduce ROS and malondialdehyde (MDA) levels, increase…”
L65-66: “…of ADPN contribute to the protection of GCs…”
L70: “that if”?
L93: “SYF exhibited a comparable pattern to atretic follicles” Comparable or higher?
L100: mean ± SD or mean ± SEM (L567)?
L105: Please delete “that”
L208-209: “…decreased mRNA expression of…”
L211-212: Please specify
L218: “are shown”
L306: “…hierarchical sequence, and only a…”
L312: “In the present study…”
L315-316: “According to the study” Your study or previous one?
L323: “Our findings are consistent with this hypothesis, as…”
L374-375: “Luti et al. [37] confirmed the…” Please delete “[37]” from the end of the sentence
L377: “is expressed”
L380: “pig” instead of “porcine”
L391: “…our and others research [40-42] that ADPN can activate autophagy. Overexpression…”
L396: “…have reported the role of…”
L405-406: “…of GCs, promoting inflammation and suppressing the immune response.”
L417: “Our aforementioned results…”
L419: “…studies, it is suggested that…”
L448: How many animals were used?
L450: “bleeding”? Do you mean “sacrificing”?
L461: “collected” instead of “taken”
L491: “…were overnightly incubated at 4°C with…”
L496: Please delete “overnight at 4°C”
L510: "...Gilbert et al. [62], and digested and cultured according to the method of Kang et al. [63].”
Comments on the Quality of English LanguageMinor editing of English language required
Reviewer 2 Report
Comments and Suggestions for Authors
The manuscript entitled “ADPN Regulates Oxidative Stress-induced Follicular Atresia in Geese by Modulating Granulosa Cells Apoptosis and Autophagy” is well-written and well-structured by the authors, who demonstrate considerable scientific rigor in their approach and conduct of the investigation.
I do not find any particular suggestions of a conceptual nature in the development of the experimental part, but some very important ones of a formal nature are definitely worth considering to improve the manuscript.
Figures 3, 4 and 5 are at the limit of comprehension because of their small size. They should be brought to at least the size of Figure 1 if not even larger.
The same consideration applies to the next figures 7 through 13.
In particular, Figures 12 and 13 are absolutely unintelligible so they should definitely be increased in size or even be divided into several separate figures. I do not know if the journal guidelines allow this, otherwise they could be placed within the supplementary material while making them well interpretable by the reader.
The discussion is very detailed and delves into the various aspects arising from the many results obtained from the survey, however, some comparisons with recent literature might be important to complement what the Authors have already covered.
Reviewer 3 Report
Comments and Suggestions for Authors
The authors have found the adiponectin overexpression results in the regulation of the hormone secretion pathway but the authors did not show or check the levels of reproductive hormones in geese?
The domestic geese experience an annual cycle of reproductive quiescence and recrudescence. Did the authors follow the experiments in their reproductive cycle? Justify.
As the adiponectin or ADPN agonist works by activating its two well-known receptors adipoR1/R2. The authors did not measure the expression of any of the receptors in this manuscript in either mRNA or protein level. The authors did not cited the recent literature of adiponectin in reproductive cycles and oxidative stress parameters (PMID: 29669464, 30471430, 33706964, 29908833).
Focus on stress-related pathways in the RNAseq data and correlate with the genes with adiponectin to see the changes.
Minor spelling checks should be performed.
Round 2
Reviewer 3 Report
Comments and Suggestions for Authors
Thank you for addressing all the comments raised and improvising in the manuscript as needed to improve the quality of the manuscript.
The authors must cite before publication in IJMS PMID: 29669464 (Adiponectin in female reproductive function).
Author Response
Dear reviewer:
Thank you for your insightful suggestion. As you pointed out,We have added this literature (PMID: 29669464) citation and marked it with highlighting. Thank you again for your patience in reviewing it.